# Behavior From the Void: Unsupervised Active Pre-Training

**Hao Liu**
UC Berkeley
hao.liu@cs.berkeley.edu

**Pieter Abbeel**
UC Berkeley
pabbeel@cs.berkeley.edu

## Abstract

We introduce a new unsupervised pre-training method for reinforcement learning called APT, which stands for Active Pre-Training. APT learns behaviors and representations by actively searching for novel states in reward-free environments. The key novel idea is to explore the environment by maximizing a non-parametric entropy computed in an abstract representation space, which avoids challenging density modeling and consequently allows our approach to scale much better in environments that have high-dimensional observations (e.g., image observations). We empirically evaluate APT by exposing task-specific reward after a long unsupervised pre-training phase. In Atari games, APT achieves human-level performance on 12 games and obtains highly competitive performance compared to canonical fully supervised RL algorithms. On DMControl suite, APT beats all baselines in terms of asymptotic performance and data efficiency and dramatically improves performance on tasks that are extremely difficult to train from scratch.

## 1 Introduction

Reinforcement learning (RL) provides a general framework for solving challenging sequential decision-making problems. When combined with function approximation, it has achieved remarkable success in advancing the frontier of AI technologies. These landmarks include outperforming humans in computer games [40, 51, 64, 5] and solving complex robotic control tasks [3, 1]. Despite these successes, they have to train from scratch to maximize extrinsic reward for every encountered task. This is in sharp contrast with how intelligent creatures quickly adapt to new tasks by leveraging previously acquired behaviors. Unsupervised pre-training, a framework that trains models without expert supervision, has obtained promising results in computer vision [43, 23, 14] and natural language modeling [63, 16, 11]. The learned representation, when fine-tuned on the downstream tasks, can solve them efficiently in a few-shot manner. With the models and datasets growing, performance continues to improve predictably according to scaling laws.

Driven by the significance of massive unlabeled data, we consider an analogy setting of unsupervised pre-training in computer vision where labels are removed during training. The goal of pre-training is to have data efficient adaptation for some downstream task defined in the form of rewards. In RL with unsupervised pre-training, the agent is allowed to train for a long period without access to environment reward, and then only gets exposed to the reward during testing. We first test an array of existing methods for unsupervised pre-training to identity which gaps and challenges exist, we evaluate count-based bonus [10], which encourages the agent to visit novel states. We apply count-based bonus to DrQ [33] which is current state-of-the-art RL for training from pixels. We also evaluate ImageNet pre-trained representations. The results are shown in Figure 1. We can see that count-based bonus fails to outperform train DrQ from scratch. We hypothesize that the ineffectiveness stems from density modeling at the pixel level being difficult. ImageNet pre-training does not outperform training from scratch either, which has also been shown in previous research

35th Conference on Neural Information Processing Systems (NeurIPS 2021).

in real world robotics [29]. We believe the reason is that neither of existing methods can provide enough diverse data. Count-based exploration faces the difficult of estimating high dimensional data density while ImageNet dataset is out-of-distribution for DMControl.

To address the issue of obtaining diverse data for RL with unsupervised pre-training, we propose to actively collect novel data by exploring unknown areas in the task-agnostic environment. The underlying intuition is that a general exploration strategy has to visit, with high probability, any state where the agent might be rewarded in a subsequent RL task. Concretely, our approach relies on the entropy maximization principle [27, 53]. Our hope is that by doing so, the learned behavior and representation can be trained on the whole environment while being as task agnostic as possible. Since entropy maximization in high dimensional state space is intractable as an oracle density model is not available, we resort to the particle-based entropy estimator [55, 8]. This estimator is nonparametric and asymptotically unbiased. The key idea is computing the average of the Euclidean distance of each particle to its nearest neighbors for a set of samples. We consider an abstract representation space in order to make the distance meaningful. To learn such a representation

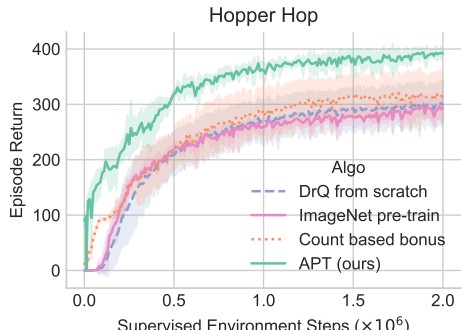

**Figure 1:** Comparison of state-of-the-art pixel-based RL with unsupervised pre-training. APT (ours) and count-based bonus (both based on DrQ [33]) are trained for a long unsupervised period (5M environment steps) without access to environment reward, and then gets exposure to the environment reward during testing. APT significantly outperform training DrQ from scratch, count-based bonus, and ImageNet pre-trained model.

space, we adapt the idea of contrastive representation learning [14] to encode image observations to a lower dimensional space. Building upon this insight, we propose Unsupervised Active Pre-Training (APT) since the agent is encouraged to actively explore and leverage the experience to learn behavior.

Our approach can be applied to a wide-range of existing RL algorithms. In this paper we consider applying our approach to DrQ [33] which is a state-of-the-art visual RL algorithm. On the Atari 26 games subset, APT significantly improves DrQ's data-efficiency, achieving 54% relative improvement. On the full suite of Atari 57 games [40], APT significantly outperforms prior state-of-the-art, achieving a median human-normalized score $3\times$ higher than the highest score achieved by prior unsupervised RL methods and DQN. On DeepMind control suite, APT beats DrQ and unsupervised RL in terms of asymptotic performance and data efficiency and solving tasks that are extremely difficult to train from scratch. The contributions of our paper can be summarized as: (i) We propose a new approach for unsupervised pre-training for visual RL based a nonparametric particle-based entropy maximization. (ii) We show that our pre-training method significantly improves data efficiency of solving downstream tasks on DMControl and Atari suite.

## 2 Problem Setting

**Reinforcement Learning (RL)**  An agent interacts with its uncertain environment over discrete timesteps and collects reward per action, modeled as a Markov Decision Process (MDP) [48], defined by $\langle \mathcal{S}, \mathcal{A}, T, \rho_0, r, \gamma \rangle$ where $\mathcal{S} \subseteq \mathbb{R}^{n_\mathcal{S}}$ is a set of $n_\mathcal{S}$-dimensional states, $\mathcal{A} \subseteq \mathbb{R}^{n_\mathcal{A}}$ is a set of $n_\mathcal{A}$-dimensional actions, $T : \mathcal{S} \times \mathcal{A} \times \mathcal{S} \to [0, 1]$ is the state transition probability distribution. $\rho_0 : \mathcal{S} \to [0, 1]$ is the distribution over initial states, $r : \mathcal{S} \times \mathcal{A} \to \mathbb{R}$ is the reward function, and $\gamma \in [0, 1)$ is the discount factor. At environment state $s \in \mathcal{S}$, the agent take actions $a \in \mathcal{A}$, in the (unknown) environment dynamics defined by the transition probability $T(s'|s, a)$, and the reward function yields a reward immediately following the action $a_t$ performed in state $s_t$. We define the discounted return $G(s_t, a_t) = \sum_{l=0}^{\infty} \gamma^l r(s_{t+l}, a_{t+l})$ as the discounted sum of future rewards collected by the agent. In value-based reinforcement learning, the agent learns an estimate of the expected discounted return, a.k.a, state-action value function $Q^\pi(s_t, a_t) = \mathbb{E}_{s_{t+1}, a_{t+1}, \dots} \left[ \sum_{l=0}^{\infty} \gamma^l r(s_{t+l}, a_{t+l}) \right]$. A common way of deriving a new policy from a state-action value function is to act $\epsilon$-greedily with respect to the action values (discrete) or to use policy gradient to maximize the value function (continuous).

**Unsupervised Pre-Training RL**  In pretrained RL, the agent is trained in a reward-free MDP $\langle \mathcal{S}, \mathcal{S}_0, \mathcal{A}, T, \mathcal{G} \rangle$ for a long period followed by a short testing period with environment rewards $\mathbb{R}$ provided. The goal is to learn a pretrained agent that can quickly adapt to testing tasks defined

by rewards to maximize the sum of expected future rewards in a zero-shot or few-shot manner. This is also known as the two phases learning in unsupervised pretraining RL [20]. The current state-of-the-art methods maximize the mutual information ($I$) between policy-conditioning variable ($w$) and the behavior induced by the policy in terms of state visitation ($s$).

$$\max I(s; w) = \max H(w) - H(w|s),$$

where $w$ is sampled from a fixed distribution in practice as in DIAYN [17] and VISR [20]. The objective can then be simplified as $\max -H(w|s)$. Due to it being intractable to directly maximize this negative conditional entropy, prior work propose to maximize the variational lower bound of the negative conditional entropy instead [7]. The training then amounts to learning a posterior of task variable conditioning on states $q(w|s)$.

$$-H(w|s) \geq \mathbb{E}_{s,w} \left[ \log q(w|s) \right].$$

Despite successful results in learning meaningful behaviors from reward-free interactions [e.g. 41, 18, 26, 17, 20], these methods suffer from insufficient exploration because they contain no explicit exploration.

Another category considers the alternative direction of maximizing the mutual information [12].

$$\max I(s; w) = \max H(s) - H(s|w).$$

This intractable quantity can be similarly lowered bound by a variational approximation [7].

$$I(s; w) \geq \mathbb{E}_{s,w} \left[ q_\theta(s|w) \right] - \mathbb{E}_s \left[ \log p(s) \right],$$

where $\mathbb{E}_s \left[ \log p(s) \right]$ can then be approximated by a posterior of state given task variables $\mathbb{E}_s \left[ \log p(s) \right] \approx \mathbb{E}_{s,w} \left[ \log q(s|w) \right]$. Despite their successes, this category of methods do not explore sufficiently since the agent receives larger rewards for visiting known states than discovering new ones as theoretically and empirically evidenced by Campos et al. [12]. In addition, they have only been shown to work from explicit state-representations and it remains unclear how to modify to learning from pixels.

In the next section, we introduce a new nonparametric unsupervised pre-training method for RL which addresses these issues and outperforms prior state-of-the-arts on challenging visual-domain RL benchmarks.

## 3 Unsupervised Active Pre-Training for RL

We want to incentivize the agent with a reward $r_t$ to maximize entropy in an abstract representation space. Prior work on maximizing entropy relies on estimating density of states which is challenging and non-trivial, instead, we take a two-step approach. First, we learn a mapping $f_\theta : R^{n_S} \rightarrow R^{n_Z}$ that maps state space to an abstract representation space first. Then, we propose a particle-based nonparametric approach to maximize the entropy by deploying state-of-the-art RL algorithms.

We introduce how to maximize entropy via particle-based approximation in Section 3.1, and describe how to learn representation from states in Section 3.2

### 3.1 Particle-Based Entropy Maximization

Our entropy maximization objective is built upon the nonparametric particle-based entropy estimator proposed by Singh et al. [55] and Beirlant [8] and has has been widely studied in statistics [28]. Its key idea is to measure the sparsity of the distribution by considering the distance between each sampled data point and its $k$ nearest neighbors. Concretely, assuming we have number of $n$ data points $\{z_i\}_{i=1}^n$ from some unknown distribution, the particle-based approximation can be written as

$$H_{\text{particle}}(z) = -\frac{1}{n} \sum_{i=1}^n \log \frac{k}{n v_i^k} + b(k) \propto \sum_{i=1}^n \log v_i^k, \tag{1}$$

where $b(k)$ is a bias correction term that only depends on the hyperparameter $k$, and $v_i^k$ is the volume of the hypersphere of radius $\|z_i - z_i^{(k)}\|$ between $z_i$ and its $k$-th nearest neighbor $z_i^{(k)}$. $\|\cdot\|$ is the Euclidean distance.

$$v_i^k = \frac{\|z_i - z_i^{(k)}\|^{n_Z} \cdot \pi^{n_Z/2}}{\Gamma(n_Z/2 + 1)}, \tag{2}$$

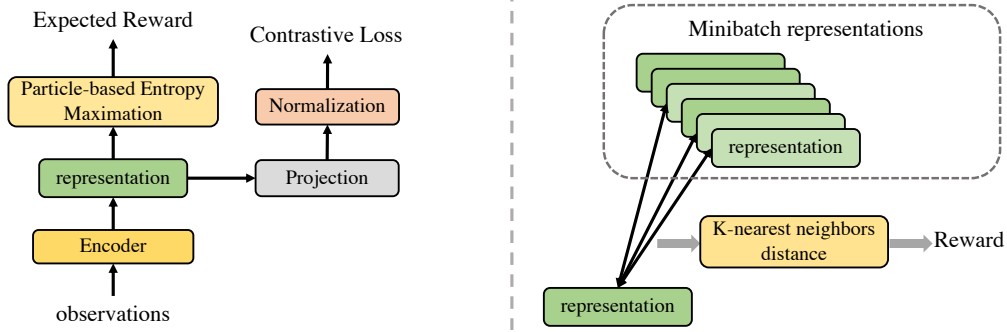

**Figure 2:** Diagram of the proposed method APT. On the left shows the objective of APT, which is to maximize the expected reward and minimize the contrastive loss. The contrastive loss learns an abstract representation from observations induced by the policy. We propose a particle-based entropy maximization based reward function such that we can deploy state-of-the-art RL methods to maximize entropy in an abstraction space of the induced by the policy. On the right shows the idea of our particle-based entropy, which measures the distance between each data point and its k nearest neighbors.

where $\Gamma$ is the gamma function. Intuitively, $v_i^k$ reflects the sparsity around each particle and equation (1) is proportional to the average of the volumes around each particle.

By substituting equation (2) into equation (1), we can simplify the particle-based entropy estimation as a sum of the log of the distance between each particle and its $k$-th nearest neighbor.

$$H_{\text{particle}}(z) \propto \sum_{i=1}^{n} \log \|z_i - z_i^{(k)}\|^{n_z}. \tag{3}$$

Rather than using equation (3) as the entropy estimation, we find averaging the distance over all $k$ nearest neighbors leads to a more robust and stable result, yielding our estimation of the entropy.

$$H_{\text{particle}}(z) := \sum_{i=1}^{n} \log \left( c + \frac{1}{k} \sum_{z_i^{(j)} \in N_k(z_i)} \|z_i - z_i^{(j)}\|^{n_z} \right), \tag{4}$$

where $N_k(\cdot)$ denotes the $k$ nearest neighbors around a particle, $c$ is a constant for numerical stability (fixed to 1 in all our experiments).

We can view the particle-based entropy in equation (4) as an expected reward with the reward function being $r(z_i) = \log \left( c + \frac{1}{k} \sum_{z_i^{(j)} \in N_k(z_i)} \|z_i - z_i^{(j)}\|^{n_z} \right)$ for each particle $z_i$. This makes it possible to deploy RL algorithms to maximize entropy, concretely, for a batch of transitions $\{(s, a, s')\}$ sampled from the replay buffer. We consider the representation of each $s'$ as a particle in the representation space and the reward function for each transition is given by

$$r(s, a, s') = \log \left( c + \frac{1}{k} \sum_{z^{(j)} \in N_k(z = f_\theta(s))} \|f_\theta(s) - z^{(j)}\|^{n_z} \right) \tag{5}$$

In order to keep the rewards on a consistent scale, we normalize the intrinsic reward by dividing it by a running estimate of the mean of the intrinsic reward. See Figure 2 for illustration of the formulation.

## 3.2 Learning Contrastive Representations

Our aforementioned entropy maximization is modular of the representation learning method we choose to use, the representation learning part can be swapped out for different methods if necessary. However, for entropy maximization to work, the representation needs to contain a compressed representation of the state. Recent work, CURL [35], ATC [56] and SPR [52], show contrastive learning (with data augmentation) helps learn meaningful representations in RL. We choose contrastive representation learning since it maximally distinguishes an observation $s_{t_1}$ from alternative observations $s_{t_2}$ according to certain distance metric in representation space, we hypothesize is helpful for learning meaningful representations for our nearest neighbors based entropy maximization. Our contrastive

learning is based on the contrastive loss from SimCLR [14], chosen for its simplicity. We also use the same set of image augmentations as in DrQ [33] consisting of small random shifts and color jitter. Concretely, we randomly sample a batch of states (images) from the replay buffer $\{s_i\}_{i=1}^n$. For each state $s_i$, we apply random data augmentation and obtain two randomly augmented views of the same state, denoted as key $s_i^k = \text{aug}(s_i)$ and query $s_i^v = \text{aug}(s_i)$. The augmented observations are encoded into a small latent space using the encoder $z = f_\theta(\cdot)$ followed by a deterministic projection $h_\phi(\cdot)$ where a contrastive loss is applied. The goal of contrastive learning is to ensure that after the encoder and projection, $s_i^k$ is relatively more close to $s_i^v$ than any of the data points $\{s_j^k, s_j^v\}_{j=1, j\neq i}^n$.

$$\min_{\theta,\phi} -\frac{1}{2n} \sum_{i=1}^n \left[ \log \frac{\exp(h_\phi(f_\theta(s_i^k))^T h_\phi(f_\theta(s_i^v)))}{\sum_{i=1}^n \mathbb{I}_{[j\neq i]}(\exp(h_\phi(f_\theta(s_i^k))^T h_\phi(f_\theta(s_j^k))) + \exp(h_\phi(f_\theta(s_i^k))^T h_\phi(f_\theta(s_j^v))))} \right].$$

Following DrQ, the representation encoder $f_\theta(\cdot)$ is implemented by the convolutional residual network followed by a fully-connected layer, a `LayerNorm` and a `Tanh` non-linearity. We decrease the output dimension of the fully-connected layer after the convnet from 50 to 15. We find it helps to use spectral normalization [39] to normalize the weights and use ELU [15] as the non-linearity in between convolutional layers.

Table 1 positions our new approach with respect to existing ones. Figure 2 shows the resulting model. Training proceeds as in other algorithms maximizing extrinsic reward: by learning neural encoder $f$ and computing intrinsic reward $r$ and then trying to maximize this intrinsic return by training the policy. Algorithm 1 shows the pseudo-code of APT, we highlight the changes from DrQ to APT in color.

---

**Algorithm 1:** Training APT

---

Randomly Initialize $f$ encoder
Randomly Initialize $\pi$ and $Q$ networks
**for** $e := 1, \infty$ **do**
  **for** $t := 1, T$ **do**
    Receive observation $s_t$ from environment
    Take action $a_t \sim \pi(\cdot|s_t)$, receive observation $s_{t+1}$ and $r_t$ from environment
    $\mathcal{D} \leftarrow \mathcal{D} \cup (s_t, a_t, r_t, s_t')$
    $\{(s_i, a_i, r_i, s_i')\}_{i=1}^N \sim \mathcal{D}$                    // sample a mini batch
    Train neural encoder $f$ on mini batch                    // representation learning
    **for** each $i = 1..N$ **do**
      $a_i' \sim \pi(\cdot|s_i')$
      $\hat{Q}_i = Q_{\theta'}(s_i', a_i')$
      Compute $r_{\text{APT}}$ with equation (5)                    // particle-based entropy reward
      $y_i \leftarrow r_{\text{APT}} + \gamma \hat{Q}_i$
    **end**
    $loss_Q = \sum_i (Q(s_i, a_i) - y_i)^2$
    Gradient descent step on $Q$ and $\pi$                    // standard actor-critic
  **end**
**end**

---

**Table 1:** Methods for pre-training RL in reward-free setting. Exploration: the method can explore efficiently. Visual: the method works well in visual RL. Off-policy: the method is compatible with off-policy RL optimization. $^\star$ means only in state-based RL. c(s) is count-based bonus. $\psi(s, a)$: successor feature, $\phi(s)$: state representation.

| Algorithm | Objective | Visual | Exploration | Off-policy | Pre-Trained model |
|-----------|-----------|--------|-------------|------------|-------------------|
| MaxEnt [22] | $\max \text{H}(s)$ | ✗ | ✓$^\star$ | ✗ | $\pi(a\|s)$ |
| CBB [10] | $\max \mathbb{E}_s[c(s)]$ | ✗ | ✓ | ✓ | $\pi(a\|s)$ |
| MEPOL [42] | $\max \text{H}(s)$ | ✗ | ✓$^\star$ | ✗ | $\pi(a\|s)$ |
| VISR [20] | $\max -\text{H}(z\|s)$ | ✓ | ✗ | ✓ | $\psi(s, z), \phi(s)$ |
| DIAYN [17] | $\max -\text{H}(z\|s) + \text{H}(a\|z, s)$ | ✗ | ✓$^\star$ | ✓ | $\pi(a\|s, z)$ |
| DADS [54] | $\max \text{H}(s) - \text{H}(s\|z)$ | ✗ | ✗ | ✓ | $\pi(a\|s, z), q(s'\|s, z)$ |
| EDL [12] | $\max \text{H}(s) - \text{H}(s\|z)$ | ✗ | ✓$^\star$ | ✓ | $\pi(a\|s, z)$ |
| APT | $\max \text{H}(s)$ | ✓ | ✓ | ✓ | $\pi(a\|s), Q(s, a)$ |

## 4 Related Work

**State Space Entropy Maximization**. Maximizing entropy of policy has been widely studied in RL, from inverse RL [69] to optimal control [59, 60, 49] and actor-critic [19]. State space entropy maximization has been recently used as an exploration method by estimating density of states and maximizing entropy [22]. In Hazan et al. [22] they present provably efficient exploration algorithms under certain conditions. VAE [32] based entropy estimation has been deployed in lower dimensional observation space [36]. However, due to the difficulty of estimating density in high dimensional space such as Atari games, such parametric exploration methods struggle to work in more challenging visual domains. In contrast, our work turns to particle based entropy maximization in a contrastive representation space. Maximizing particle-based entropy has been shown to improved data efficiency in state-based RL as in MEPOL [42]. However, MEPOL's entropy estimation depends on importance sampling and the optimization based on on-policy RL algorithms, hindering further applications to challenging visual domains. MEPOL also assumes having access to the semantic information of the state, making it infeasible and not obvious how to modify it to work from pixels. In contrast, our method is compatible with deploying state-of-the-art off-policy RL and representation learning algorithms to maximize entropy. Nonparametric entropy maximization has been studied in goal conditioned RL [66]. Pitis et al. [47] proposes maximizing entropy of achieved goals and demonstrates significantly improved success rates in long horizon goal conditioned tasks. The work by Badia et al. [6] also considers k-nearest neighbor based count bonus to encourage exploration, yielding improved performance in Atari games. K-nearest neighbor based exploration is shown to improve exploration and data efficiency in model-based RL [57]. Concurrently, it has been shown to be an effective unsupervised pre-training objective for transferring learning in RL [13], their large scale experiments further demonstrate the effectiveness of unsupervised pre-training.

**Data Efficient RL**. To improve upon the sample efficiency of deep RL methods, various methods have been proposed: Kaiser et al. [30] introduce a model-based agent (SimPLe) and show that it compares favorably to standard RL algorithms when data is limited. Hessel et al. [25], Kielak [31], van Hasselt et al. [61] show combining existing RL algorithms (Rainbow) can boost data efficiency. Data augmentation has also been shown to be effective for improving data efficiency in vision-based RL [34, 33]. Temporal contrastive learning combined with model-based learning has been shown to boost data efficiency [52]. Combining contrastive loss with RL has been shown to improve data efficiency in CPC [24] despite only marginal gains. CURL [35] show substantial data-efficiency gains while follow-up results from Kostrikov et al. [33] suggest that most of the benefits come from its use of image augmentation. Contrastive loss has been shown to learn useful pretrained representations when training on expert demonstration [56], however in our work the agent has to explore the world itself and exploit collect experience.

**Unsupervised Pre-Training RL**. A number of recent works have sought to improve reinforcement learning via the addition of an unsupervised pretraining stage, in which the agent improves its representations prior to beginning learning on the target task. One common approach has been to allow the agent a period of fully-unsupervised interaction with the environment during which the agent is trained to learn a set of skills associated with different paths through the environment, as in DIAYN [17], Proto-RL [67], MUSIC [68], APS [37], and VISR [20]. Others have proposed to use self-supervised objectives to generate intrinsic rewards encouraging agents to visit new states, e.g., Pathak et al. [46] use the disagreement between an ensemble of latent-space dynamics models. However, our work is trained to maximize the entropy of the states induced by the policy. By visiting any state where the agent might be rewarded in a subsequent RL task, our work performs better or comparably well as other more complex and specialized state-of-the-art methods.

## 5 Results

We test APT in DeepMind Control Suite [DMControl; 58] and the Atari suite [9]. During the the long period of pre-training with environment rewards removed, we use DrQ to maximize the entropy maximization reward defined in equation (5). The pre-trained value function $Q(s, a)$ is fine-tuned to maximize task specific reward after being exposing to environment rewards during testing period. For our DeepMind control suite and Atari games experiments, we largely follow DrQ, except we perform two gradient steps per environment step instead of one. Our ablation studies confirm that

these changes are not themselves responsible for our performance. Kornia [50] is used for efficient GPU-based data augmentations. Our model is implemented in Numpy [21] and PyTorch [45].

**APT outperforms prior from scratch SOTA RL on DMControl**. We evaluate the performance of different methods by computing the average success rate and episodic return at the end of training.

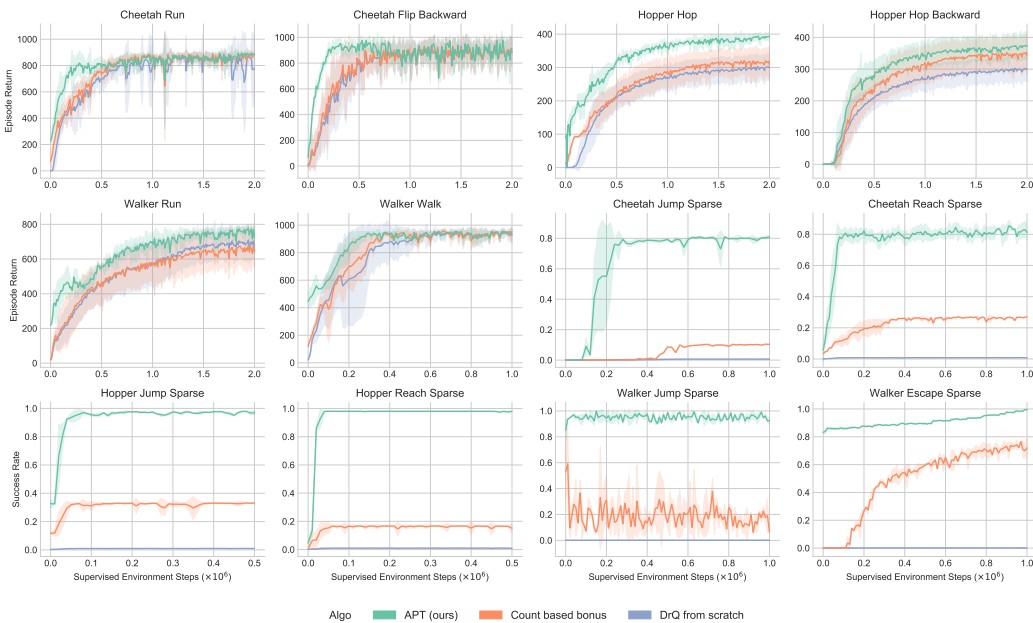

**Figure 3:** Results of different methods in environments from DMControl. All curves are the average of three runs with different seeds, and the shaded areas are standard errors of the mean.

The agent is allowed a long unsupervised pre-training phase (5M steps), followed by a short test phase exposing to downstream reward, during which the pre-trained model is fine-tuned. We follow the evaluation setting of DrQ and test APT on a subset of DMControl suite, which includes training Walker, Cheetah, Hopper for various locomotion tasks. Models are pre-trained on Cheetah, Hopper, and Walker, and subsequently fine-tuned on respective downstream tasks. We additionally design more challenging sparse reward tasks where the robot is required to accomplish tasks guided only by sparse feedback signal. The reason we opted to design new sparse reward tasks is to have more diverse downstream tasks. As far as we know, there is only one Cartpole Swingup Sparse that is a CartPole based sparse reward task. Due to its 2D nature being quite limited, we eventually decided to design distinguishable downstream tasks based on a little bit more complex environment, e.g. Hopper Jump etc. The details of the tasks are included in the supplementary material.

The learning process of RL agents becomes highly inefficient in sparse supervision tasks when relying on standard exploration techniques. This issue can be alleviated by introducing intrinsic motivation, $i.e.$, denser reward signals that can be automatically computed, one approach that works well in high dimensional setting is count-based exploration [38, 44, 38].

The results are presented in Figure 3, APT significantly outperforms SOTA training from scratch (DrQ from scratch) and SOTA exploration method (count-based bonus) on every task. With only a few number of environment interactions, APT quickly adapt to downstream tasks and achieves higher return much more quicker than prior state-of-the-art RL algorithms. Notably, on the sparse reward tasks that are extremely difficult for training from scratch, APT yields significantly higher data efficiency and asymptotic performance.

**APT outperforms from scratch SOTA RL in Atari**. We test APT on the sample-efficient Atari setting [30, 61] which consists of the 26 easiest games in the Atari suite (as judged by above random performance for their algorithm).

We follow the evaluation setting in VISR, agents are allowed a long unsupervised training phase (250M steps) without access to rewards, followed by a short test phase with rewards. The test phase contains 100K environment steps – equivalent to 400k frames, or just under two hours – compared to

the typical standard of 500M environment steps, or roughly 39 days of experience. We normalize the episodic return with respect to expert human scores to account for different scales of scores in each game, as done in previous works. The human-normalized scores (HNS) of an agent on a game is calculated as $\frac{\text{agent score} - \text{random score}}{\text{human score} - \text{random score}}$ and aggregated across games by mean or median.

A full list of scores and aggregate metrics on the Atari 26 subset is presented in Table 2. The results on the full 57 Atari games suite is presented in supplementary material. For consistency with previous works, we report human and random scores from [25]. In the data-limited setting, APT achieves super-human performance on eight games and achieves scores higher than previous state-of-the-arts. In the full suite setting, APT achieves super-human performance on 15 games, compared to a maximum of 12 for any previous methods and achieves scores significantly higher than any previous methods.

**Table 2:** Performance of different methods on the 26 Atari games considered by [30] after 100K environment steps. The results are recorded at the end of training and averaged over 10 random seeds for APT. APT outperforms prior methods on all aggregate metrics, and exceeds expert human performance on 7 out of 26 games while using a similar amount of experience. Prior work has reported different numbers for some of the baselines, particularly SimPLe and DQN. To be rigorous, we pick the best number for each game across the tables reported in van Hasselt et al. [61] and Kielak [31].

| Game | Random | Human | SimPLe | DER | CURL | DrQ | SPR | VISR | APT (ours) |
|---|---|---|---|---|---|---|---|---|---|
| Alien | 227.8 | 7127.7 | 616,9 | 739.9 | 558.2 | 771.2 | 801.5 | 364.4 | **2614.8** |
| Amidar | 5.8 | 1719.5 | 88.0 | 188.6 | 142.1 | 102.8 | 176.3 | 186.0 | **211.5** |
| Assault | 222.4 | 742.0 | 527.2 | 431.2 | 600.6 | 452.4 | 571.0 | **12091.1** | 891.5 |
| Asterix | 210.0 | 8503.3 | 1128.3 | 470.8 | 734.5 | 603.5 | 977.8 | **6216.7** | 185.5 |
| Bank Heist | 14.2 | 753.1 | 34.2 | 51.0 | 131.6 | 168.9 | 380.9 | 71.3 | **416.7** |
| BattleZone | 2360.0 | 37187.5 | 5184.4 | 10124.6 | 14870.0 | 12954.0 | **16651.0** | 7072.7 | 7065.1 |
| Boxing | 0.1 | 12.1 | 9.1 | 0.2 | 1.2 | 6.0 | **35.8** | 13.4 | 21.3 |
| Breakout | 1.7 | 30.5 | 16.4 | 1.9 | 4.9 | 16.1 | 17.1 | **17.9** | 10.9 |
| ChopperCommand | 811.0 | 7387.8 | **1246.9** | 861.8 | 1058.5 | 780.3 | 974.8 | 800.8 | 317.0 |
| Crazy Climber | 10780.5 | 23829.4 | **62583.6** | 16185.2 | 12146.5 | 20516.5 | 42923.6 | 49373.9 | 44128.0 |
| Demon Attack | 107805 | 35829.4 | 62583.6 | 16185.3 | 12146.5 | 20516.5 | 42923.6 | **8994.9** | 5071.8 |
| Freeway | 0.0 | 29.6 | 20.3 | 27.9 | 26.7 | 9.8 | 24.4 | -12.1 | **29.9** |
| Frostbite | 65.2 | 4334.7 | 254.7 | 866.8 | 1181.3 | 331.1 | **1821.5** | 230.9 | 1796.1 |
| Gopher | 257.6 | 2412.5 | 771.0 | 349.5 | 669.3 | 636.3 | 715.2 | 498.6 | **2590.4** |
| Hero | 1027.0 | 30826.4 | 2656.6 | 6857.0 | 6279.3 | 3736.3 | **7019.2** | 663.5 | 6789.1 |
| Jamesbond | 29.0 | 302.8 | 125.3 | 301.6 | 471.0 | 236.0 | 365.4 | **484.4** | 356.1 |
| Kangaroo | 52.0 | 3035.0 | 323.1 | 779.3 | 872.5 | 940.6 | **3276.4** | 1761.9 | 412.0 |
| Krull | 1598.0 | 2665.5 | **4539.9** | 2851.5 | 4229.6 | 4018.1 | 2688.9 | 3142.5 | 2312.0 |
| Kung Fu Master | 258.5 | 22736.3 | 17257.2 | 14346.1 | 14307.8 | 9111.0 | 13192.7 | 16754.9 | **17357.0** |
| Ms Pacman | 307.3 | 6951.6 | 1480.0 | 1204.1 | 1465.5 | 960.5 | 1313.2 | 558.5 | **2827.1** |
| Pong | -20.7 | 14.6 | **12.8** | -19.3 | -16.5 | -8.5 | -5.9 | -26.2 | -8.0 |
| Private Eye | 24.9 | 69571.3 | 58.3 | 97.8 | 218.4 | -13.6 | **124.0** | 98.3 | 96.1 |
| Qbert | 163.9 | 13455.0 | 1288.8 | 1152.9 | 1042.4 | 854.4 | 669.1 | 666.3 | **17671.2** |
| Road Runner | 11.5 | 7845.0 | 5640.6 | 9600.0 | 5661.0 | 8895.1 | **14220.5** | 6146.7 | 4782.1 |
| Seaquest | 68.4 | 42054.7 | 683.3 | 354.1 | 384.5 | 301.2 | 583.1 | 706.6 | **2116.7** |
| Up N Down | 533.4 | 11693.2 | 3350.3 | 2877.4 | 2955.2 | 3180.8 | 28138.5 | 10037.6 | 8289.4 |
| Mean HNS | 0.000 | 1.000 | 44.3 | 28.5 | 38.1 | 35.7 | **70.4** | 64.31 | 69.55 |
| Median HNS | 0.000 | 1.000 | 14.4 | 16.1 | 17.5 | 26.8 | 41.5 | 12.36 | **47.50** |
| # Superhuman | 0 | N/A | 2 | 2 | 2 | 2 | **7** | 6 | **7** |

Unsupervised pre-training on top of DrQ leads a significant increase in performance(a 54% increase in median score, a 73% increase in mean score, and 5 more games with human-level performance), surpassing DQN which trained on hundreds of millions of sampling steps.

Compared with SPR [52] which is a recent state-of-the-art model-based data-efficient algorithm, APT achieves comparable mean and median scores. The SPR is based on Rainbow which combines more advances than DrQ which is significantly simpler. While the representation of SPR is also learned by contrastive learning, it trains a model-based dynamic to predict its own latent state representations multiple steps into the future. This temporal representation learning, as illustrated in the SPR paper, contributes to its impressive results compared with standard contrastive representation learning. We believe that it is possible to combine temporal contrastive representation learning of SPR with the effective nonparametric entropy maximization of APT, which is an interesting future direction.

**APT outperforms prior unsupervised RL**. Despite there being many different proposed unsupervised RL methods, their successes are only demonstrated in simple state based environments. Prior works train the agent for a period of fully-unsupervised interaction with the environment, during which the agent is trained to learn a set of skills associated with different paths through the environ-

ment, as in DIAYN [17] and VIC [18], or to maximize the diversity of the states it encounters, as in MEPOL [42] and Hazan et al. [22]. Until recently, VISR [20] achieves improved results in Atari games using pixels as input based using a successor feature based approach. In order to compare with prior unsupervised RL methods, we choose DIAYN due to it being based on mutual information maximization and its reported high performance in state-based RL, and MEPOL due to it being based on entropy maximization. We implement them to take pixels as input in Atari games. Our implementation was checked against publicly available code and we made a best effort attempt to tune the algorithms in Atari games. We test two variants of DIAYN and MEPOL, using or not using contrastive representation learning as in APT. In order to ensure a fair comparison, we test a variant of APT without contrastive representation learning.

The aggregated results are presented in Table 3, APT significantly outperforms prior state-based unsupervised RL algorithms DIAYN and MEPOL. Both baselines benefit from contrastive representation learning, but their scores are still significantly lower than APT's score, confirming that the effectiveness of the off-policy entropy maximization in APT. Compared with the state-of-the-art method in Atari VISR, APT achieves significantly higher median score despite having a lower mean score. From the scores breakdown presented in supplementary file, APT performs significantly better than VISR in hard exploration games, while VISR achieves higher scores in dense reward games. We attribute this to that maximizing state entropy leads to more exploratory behavior while successor features enables quicker adaptation for dense reward feedback. It is possible to combine VISR and APT to have the best of both worlds, which we leave as a future work.

**Table 3:** Evaluation in Atari games. The amount of RL interaction utilized is 100K. $Mdn$ is the median of human-normalized scores, $M$ is the mean and $> H$ is the number of games with human-level performance. CL denotes training representation encoder using contrastive learning and data augmentation. On each subset, we mark as bold the highest score.

| Algorithm | 26 Game Subset | | | Full 57 Games | | |
| --- | --- | --- | --- | --- | --- | --- |
| | Mdn | M | $>H$ | Mdn | M | $>H$ |
| CBB | 1.23 | 21.94 | 3 | – | – | – |
| MEPOL | 0.34 | 17.94 | 2 | – | – | – |
| DIAYN | 1.34 | 25.39 | 2 | 2.95 | 23.90 | 6 |
| CBB w/ CL | 1.78 | 17.34 | 2 | – | – | – |
| MEPOL w/ CL | 1.05 | 21.78 | 3 | – | – | – |
| DIAYN w/ CL | 1.76 | 28.44 | 2 | 3.28 | 25.14 | 6 |
| VISR | 9.50 | **128.07** | **7** | 6.81 | **102.31** | 11 |
| APT w/o CL | 21.23 | 28.12 | 3 | 28.65 | 41.12 | 9 |
| APT | **47.50** | 69.55 | **7** | **33.41** | 47.73 | **12** |

**Ablation study**. We conduct several ablation studies to measure the contribution of each component in our method. We test two variants of APT that use the same number of gradient steps per environment step and use the same activation function as in DrQ. Another variant of APT is based on randomly selected neighbors to compute particle-based entropy.

We also test a variant of APT that use a fixed randomly initialized encoder to study the impact of representation learning. Table 4 shows the performance of each variant of APT. Increasing gradient steps of updating value function from 1 to 2 and using ELU activation function yield higher scores. Using k-nearest neighbors is crucial to high scores, we believe the reason is randomly selected neighbors do not provide necessary incentive to explore. Using randomly initialized convolutional encoder downgrades performance significantly but still achieve higher score than DrQ, indicating our particle-based entropy maximization is robust and powerful.

**Table 4:** Scores on the 26 Atari games under consideration for variants of APT. Scores are averaged over 3 random seeds. All variants listed here use data augmentation.

| Variant | Human-Normalized Score | |
| --- | --- | --- |
| | median | mean |
| APT | **47.50** | **69.55** |
| APT w/o optim change | 41.50 | 60.10 |
| APT w/o arch change | 45.71 | 67.82 |
| APT w/ rand neighbor | 20.80 | 24.97 |
| APT w/ fixed encoder | 33.24 | 41.08 |

Contrastive learning representation has been shown to have the "uniformity on the hypersphere" property [65], this leads to the question that whether maximum entropy exploration in state space is important. To study this question, we have a variant of APT "Pos Reward APT" which receives a simple positive do not die signal but no particle-based entropy reward. We ran the experiments on MsPacman, we reduced the pretraining phase to 5M steps to reduce computation cost. The evaluation metrics are the number of ram states visited using [2] and the downstream zero shot performance on Atari game. APT visits nearly 27 times more unique ram states than "Pos Reward APT", showing that the entropy intrinsic reward is indispensable for exploration. In downstream task evaluation over 3 random seeds, "Pos Reward APT@0" achieves reward 363.7, "APT@0" achieves reward 687.1, showing that the "do not die" signal is insufficient for exploration or learning pretrained behaviors and representations.

We consider a variant of APT that re-initialize the head of pretrained actor-critic. We have run experiments in five different Atari games, as shown in Table 5, pretrained heads

**Table 5:** Scores on 5 Atari games under consideration for different variants of fine-tuning. Scores are averaged over 3 random seeds.

| Mean Reward (3 seeds) | Alien | Freeway | Qbert | Private Eye | MsPacman |
|---|---|---|---|---|---|
| APT (pretrained head) | **2614.8** | **29.9** | 17671.2 | **96.1** | **2827.1** |
| APT (random head) | 1755.0 | 15.2 | **2138.3** | 61.3 | 1724.9 |

perform better than randomly initialized heads in 4 out of 5 games. The experiments demonstrate that finetuning from a pretrained actor-critic head accelerates learning. However, we believe that which one of the two is better depends on the alignment between downstream reward and intrinsic reward. It would be interesting to study how to better leverage downstream reward to finetune the pretrained model.

## 6   Discussion

**Limitation:**   The fine-tuning strategy employed here (when combined with a value function) works best when the intrinsic and extrinsic rewards being of a similar scale. We believe the discrepancy between intrinsic reward scale and downstream reward scale possibly explain the suboptimal performance of APT in dense reward games. This is an interesting future direction to further improve APT, we hypothesize that reinitializing behaviors part (actor-critic heads) might be useful if the downstream reward scale is very different from pretraining reward scale. One of the principled ways could be adaptive normalization [62], it is an interesting future direction. One challenge of our method is the non-stationarity of the intrinsic reward, being non additive reward poses an interesting challenge for reinforcement learning methods. While our method outperforms training from scratch and prior works, we believe designing better optimization RL methods for maximizing our intrinsic reward can lead to more significant improvement.

**Conclusion:**   A new unsupervised pre-training method for RL is introduced to address reward-free pre-training for visual RL, allowing the same task-agnostic pre-trained model to successfully tackle a broad set of RL tasks. Our major contribution is introducing a practical intrinsic reward derived from particle-based entropy maximization in abstract representation space. Empirical study on DMControl suite and Atari games show our method dramatically improves performance on tasks that are extremely difficult for training from scratch. Our method achieves the results of fully supervised canonical RL algorithms using a small fraction of total samples and outperforms data-efficient supervised RL methods.

For future work, there are a few ways in which our method can be improved. The long pre-training phase in our work is computationally intensive, since the exhaustive search and exploration is of high sample complexity. One way to remedy this is by combining our method with successful model-based RL and search approaches to reduce sample complexity. Furthermore, fine-tuning the whole pre-trained model can make it prone to catastrophic forgetting. As such, it is worth studying alternative methods to leverage the pre-trained models such as keeping the pretrained model unchanged and combine it with a randomly initialized model.

## 7   Acknowledgment

This research was supported by DARPA Data-Driven Discovery of Models (D3M) program. We would like to thank Misha Laskin, Olivia Watkins, Qiyang Li, Lerrel Pinto, Kimin Lee and other members at RLL and BAIR for insightful discussion and giving constructive comments. We would also like to thank anonymous reviewers for their helpful feedback for previous versions of our work.

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
