# A  General Implementation Details

For our Atari games and DeepMind Control Suite experiments, we largely follow DrQ [33], with the following exceptions. We use three layer convolutional neural network from [40] for policy network, and the Impala architecture for neural encoder with LSTM module removed. We use the ELU nonlinearity [15] in between layers of the encoder. The number of power iterations is 5 in spectral normalization.

The convolution neural network is followed by a full-connected layer normalized by `LayerNorm` [4] and a `tanh` nonlinearity applied to the output of fully-connected layer.

The data augmentation is a simple random shift which has been shown effective in visual domain RL in DrQ [33] and RAD [34]. Specifically, the images are padded each side by 4 pixels (by repeating boundary pixels) and then select a random $84 \times 84$ crop, yielding the original image. The replay buffer size is 100K. This procedure is repeated every time an image is sampled from the replay buffer. The learning rate of contrastive learning is $0.001$, the temperature is $0.1$. The projection network is a two-layer MLP with hidden size of 128 and output size of 64. Batch size used in both RL and representation learning is 512. The pre-training phase consists of 5M environment steps on DMControl and 250M environment steps on Atari games. The evaluation is done for 125K environment steps at the end of training for 100K environment steps.

The implementation of APT can be found at `https://github.com/rll-research/url_benchmark`.

# B  Atari Details

The corresponding hyperparameters used in Atari experiments are shown in Table 7 and Table 8.

# C  DeepMind Control Suite Details

The action repeat hyperparameters are show in Table 6. The corresponding hyperparameters used in DMControl experiments are shown in Table 9 and Table 8.

**Table 6:** The action repeat hyper-parameter used for each environment.

| Environment name | Action repeat |
|---|---|
| Cheetah | 4 |
| Walker | 2 |
| Hopper | 2 |

# D  Asymptotic Behavior of Intrinsic Reward

With the intrinsic reward defined in equation (5), we can derive that the intrinsic reward decreases to 0 as more of the state space is visited, which is a favorable property for pre-training.

**Proposition 1.** *Assume the MDP is episodic and its state space is finite $\mathcal{S} \subseteq \mathbb{R}^{n_\mathcal{S}}$, the representation encoder $f_\theta : \mathbb{R}^{n_\mathcal{S}} \to \mathbb{R}^{n_z}$ is deterministic, and we have a buffer of observed states $(s_1, \ldots, s_T)$ with total sample size $T$. For an optimal policy that maximizes the intrinsic rewards defined as in equation (5) with $k \in \mathbb{N}$, we can derive the intrinsic reward is 0 in the limit of sample size $T$.*

$$\lim_{T \to \infty} r(s, a, s') = 0, \ \forall s \in \mathcal{S}.$$

While the assumption of finite state space may not be true for large complex environment like Atari games, Proposition 1 gives more insights on using this particular intrinsic reward for pre-training.

*Proof.* Since the intrinsic reward $r(s, a, s')$ defined in equation (5) depends on the $k$ nearest neighbors in latent space and the encoder $f_\theta$ is deterministic, we just need to prove the visitation count $c(s)$ of $s$ is larger than $k$ as $T$ goes infinity. We know the MDP is episodic, therefore as $T \to \infty$, all states communicate and $c(s) \to \infty$, thus we have $\lim_{T \to \infty} c(s) \geq k, \forall k \in \mathbb{N}, \forall s \in \mathcal{S}$. □

**Table 7:** Hyper-parameters in the Atari suite experiments.

| Parameter | Setting |
|---|---|
| Data augmentation | Random shifts and Intensity |
| Grey-scaling | True |
| Observation down-sampling | $84 \times 84$ |
| Frames stacked | 4 |
| Action repetitions | 4 |
| Reward clipping | $[-1, 1]$ |
| Terminal on loss of life | True |
| Max frames per episode | 108k |
| Update | Double Q |
| Dueling | True |
| Target network: update period | 1 |
| Discount factor | 0.99 |
| Minibatch size | 32 |
| RL optimizer | Adam |
| RL optimizer (pre-training): learning rate | 0.0001 |
| RL optimizer (fine-tuning): learning rate | 0.001 |
| RL optimizer: $\beta_1$ | 0.9 |
| RL optimizer: $\beta_2$ | 0.999 |
| RL optimizer: $\epsilon$ | 0.00015 |
| Max gradient norm | 10 |
| Training steps | 100k |
| Evaluation steps | 125k |
| Min replay size for sampling | 1600 |
| Memory size | Unbounded |
| Replay period every | 1 step |
| Multi-step return length | 10 |
| Q network: channels | $32, 64, 64$ |
| Q network: filter size | $8 \times 8, 4 \times 4, 3 \times 3$ |
| Q network: stride | $4, 2, 1$ |
| Q network: hidden units | 512 |
| Non-linearity | ReLU |
| Exploration | $\epsilon$-greedy |
| $\epsilon$-decay | 2500 |

**Table 8:** Hyper-parameters for Learning the Neural Encoder.

| Parameter | Setting |
|---|---|
| Value of k | search in $\{3, 5, 10\}$ |
| Temperature | 0.1 |
| Non-linearity | ELU |
| Network architecture | same as the Q network encoder (Atari) or the shared encoder (DMControl) |
| FC hidden size | 1024 |
| Output size | 5 |

**Table 9:** Hyper-parameters in the DeepMind control suite experiments.

| Parameter | Setting |
|---|---|
| Data augmentation | Random shifts |
| Frames stacked | 3 |
| Action repetitions | Table 6 |
| Replay buffer capacity | 100000 |
| Random steps (fine-tuning phase) | 1000 |
| RL minibatch size | 512 |
| Discount $\gamma$ | 0.99 |
| RL optimizer | Adam |
| RL learning rate | $10^{-3}$ |
| Contrastive Learning Temperature | 0.1 |
| Shared encoder: channels | $32, 32, 32$ |
| Shared encoder: filter size | $3 \times 3, 3 \times 3, 3 \times 3$ |
| Shared encoder: stride | $2, 2, 2, 1$ |
| Actor update frequency | 2 |
| Actor log stddev bounds | $[-10, 2]$ |
| Actor: hidden units | 1024 |
| Actor: layers | 3 |
| Critic Q-function: hidden units | 1024 |
| Critic target update frequency | 2 |
| Critic Q-function soft-update rate $\tau$ | 0.01 |
| Non-linearity | `ReLU` |

## E  DeepMind Control Suite Sparse Environments

In addition to the existing tasks in DMControl, we tested different methods on three set customized sparse reward tasks: (1) *{HalfCheetah, Hopper, Walker} Jump Sparse*: the agent receives a positive reward 1 for jumping above a given height otherwise reward is 0. (2) *{HalfCheetah, Hopper, Walker} Reach Sparse*: the agent receives positive reward 1 for reaching a given target location otherwise reward is 0. (3) *Walker Turnover Sparse*: the initial position of Walker is turned upside down, and receives reward 1 for successfully turning itself over otherwise 0. In all the considered tasks, the episode ends when the goal is reached.

## F  Scores on the full 57 Atari games

A comparison between APT and baselines on each individual Atari game is shown in Table 10. Prior work has reported different numbers for some of the baselines, particularly SimPLe and DQN. To be rigorous, we pick the best number for each game across the tables reported in van Hasselt et al. [61] and Kielak [31]. APT achieves super-human performance on 12 games, compared to a maximum of 11 for any previous methods and achieves scores significantly higher than any previous methods.

**Table 10:** Comparison of raw scores of each method on Atari games. On each subset, we mark as bold the highest score. For VISR, due to the lack of available source code, we made a best effort attempt to reproduce the algorithm.

| Game | Random | Human | VISR | APT |
|---|---|---|---|---|
| Alien | 227.8 | 7127.7 | 364.4 | **2614.8** |
| Amidar | 5.8 | 1719.5 | 186.0 | **211.5** |
| Assault | 222.4 | 742.0 | **1209.1** | 891.5 |
| Asterix | 210.0 | 8503.3 | **6216.7** | 185.5 |
| Asteroids | 7191 | 47388.7 | **4443.3** | 678.7 |
| Atlantis | 12850.0 | 29028.1 | **140542.8** | 40231.0 |
| Bank Heist | 14.2 | 753.1 | 71.3 | **416.7** |
| Battle Zone | 2360.0 | 37187.5 | **7072.7** | 7065.1 |
| Beam Rider | 363.9 | 16826.5 | 1741.9 | **3487.2** |
| Berzerk | 123.7 | 2630.4 | 490.0 | **493.4** |
| Bowling | 23.1 | 160.7 | **21.2** | -56.5 |
| Boxing | 0.1 | 12.1 | 13.4 | **21.3** |
| Breakout | 1.7 | 30.5 | **17.9** | 10.9 |
| Centipede | 2090.9 | 12017.1 | **7184.9** | 6233.9 |
| Chopper Command | 811.0 | 7387.8 | **800.8** | 317.0 |
| Crazy Climber | 10780.5 | 23829.4 | **49373.9** | 44128.0 |
| Defender | 2874.5 | 18688.9 | **15876.1** | 5927.9 |
| Demon Attack | 107805 | 35829.4 | **8994.9** | 6871.8 |
| Double Dunk | -18.6 | -16.4 | -22.6 | **-17.2** |
| Enduro | 0.0 | 860.5 | -3.1 | **-0.3** |
| Fishing Derby | -91.7 | -38.7 | -93.9 | **-5.6** |
| Freeway | 0.0 | 29.6 | -12.1 | **29.9** |
| Frostbite | 65.2 | 4334.7 | 230.9 | **1796.1** |
| Gopher | 257.6 | 2412.5 | 498.6 | **2190.4** |
| Gravitar | 173.0 | 3351.4 | 328.1 | **542.0** |
| Hero | 1027.0 | 30826.4 | 663.5 | **6789.1** |
| Ice Hockey | -11.2 | 0.9 | **-18.1** | -30.1 |
| Jamesbond | 29.0 | 302.8 | **484.4** | 356.1 |
| Kangaroo | 52.0 | 3035.0 | **1761.9** | 412.0 |
| Krull | 1598.0 | 2665.5 | **3142.5** | 2312.0 |
| Kung Fu Master | 258.5 | 22736.3 | 16754.9 | **17357.0** |
| Montezuma Revenge | 0.0 | 4753.3 | 0.0 | **0.2** |
| Ms Pacman | 307.3 | 6951.6 | 558.5 | **2527.1** |
| Name This Game | 2292.3 | 8049.0 | **2605.8** | 1387.2 |
| Phoenix | 761.4 | 7242.6 | **7162.2** | 3874.2 |
| Pitfall | -229.4 | 6463.7 | -370.8 | **-12.8** |
| Pong | -20.7 | 14.6 | -26.2 | **-8.0** |
| Private Eye | 24.9 | 69571.3 | **98.3** | 96.1 |
| Qbert | 163.9 | 13455.0 | 666.3 | **17671.2** |
| Riverraid | 1338.5 | 17118.0 | **5422.2** | 4671.0 |
| Road Runner | 11.5 | 7845.0 | **6146.7** | 4782.1 |
| Robotank | 2.2 | 11.9 | 10.0 | **13.7** |
| Seaquest | 68.4 | 42054.7 | 706.6 | **2116.7** |
| Skiing | -17098.1 | -4336.9 | **-19692.5** | -38434.1 |
| Solaris | 1236.3 | 12326.7 | **1921.5** | 841.8 |
| Space Invaders | 148.0 | 1668.7 | **9741.0** | 3687.2 |
| Star Gunner | 664.0 | 10250.0 | **25827.5** | 8717.0 |
| Surround | -10.0 | 6.5 | -15.5 | **-2.5** |
| Tennis | -23.8 | -8.3 | 0.7 | **1.2** |
| Time Pilot | 3568.0 | 5229.2 | **4503.6** | 2567.0 |
| Tutankham | 11.4 | 167.6 | 50.7 | **124.6** |
| Up N Down | 533.4 | 11693.2 | **10037.6** | 8289.4 |
| Venture | 0.0 | 1187.5 | -1.7 | **231.0** |
| Video Pinball | 0.0 | 17667.9 | **35120.3** | 2817.1 |
| Wizard Of Wor | 563.5 | 4756.5 | 853.3 | **1265.0** |
| Yars Revenge | 3092.9 | 54576.9 | **5543.5** | 1871.5 |
| Zaxxon | 32.5 | 9173.3 | 897.5 | **3231.0** |
| Mean Human-Norm'd | 0.000 | 1.000 | **68.42** | 47.73 |
| Median Human-Norm'd | 0.000 | 1.000 | 9.41 | **33.41** |
| #Superhuman | 0 | N/A | 11 | **12** |