# OpenReview forum: "Behavior From the Void: Unsupervised Active Pre-Training"
_NeurIPS.cc/2021/Conference — NeurIPS 2021 Spotlight_

### Official Review · Reviewer_Zmg7 · 2021-07-15

**Rating:** 8
**Confidence:** 4

**Summary:**

This paper proposes an unsupervised (or rather, self-supervised) training method for RL, named active pre-training. This is based on using contrastive learning from pixels, then maximizing state entropy in the abstract space using a k-nearest neighbors density estimate. The authors combine their method with DrQ, demonstrating sample efficiency and asymptotic improvements on DeepMind control suite and Atari, along with several useful ablations.

**Limitations And Societal Impact:**

Yes, the authors addressed limitations of their work in the conclusion. Specific negative societal impacts from this work are unclear (other than those associated with machine learning generally).

**Main Review:**

**Originality**: The largest limitation of this paper appears to be a relative lack of novelty. The proposed method combines two popular ideas: self-supervised contrastive learning and state entropy maximization. As the authors discuss in section 4, these ideas have been extensively explored in previous works. Indeed, even within state entropy maximization, multiple previous works have used non-parametric density estimators. Note: I would also cite Pitis et al., 2020 as another examples of state entropy maximization via non-parametric estimators. However, in that paper, they perform density estimation in the state space rather than in a latent space. Indeed, the main distinction/contribution of this paper appears to be entropy maximization within a contrastively learned latent space. This differs from previous works that have used contrastive learning purely for pre-trained representations, as well as other works that have used non-contrastively learned latent spaces for exploration. However, given that this is the primary novelty of the paper, it is somewhat unclear and poorly motivated/justified within the paper. For example, as far as I can tell, the main justification for this choice is in sentences like

*“We hypothesize that the ineffectiveness stems from density modeling at the pixel level being difficult.”*

*“We consider an abstract representation space in order to make the distance meaningful.”*

These statements are not only imprecise, they are also not rigorously analyzed empirically. The main evidence for these claims comes from performance differences between the proposed method and other (state/observation-based) exploration methods. To summarize, this paper combines two useful techniques, contrastive representation learning and state entropy maximization, showing that their combination is even more useful. However, given that this combination is only weakly motivated in the paper, one is left wondering whether this paper provides much in the way of new insights.

Pitis et al., 2020, Maximum entropy gain exploration for long horizon multi-goal reinforcement learning.


**Quality**: Overall, the paper is fairly high quality, particularly the empirical performance evaluation. The authors compare their proposed method with various visual and exploration RL algorithms on both DeepMind control suite and Atari. The number of reported environments is extensive in both domains, and the results are competitive with SOTA algorithms. I see this evaluation as the strongest point of the paper, especially in a benchmark-focused area like RL. I also found the ablations in Table 4 to be helpful for disentangling the benefits of the various components of the algorithm.


**Clarity**: With the exception of the weak motivation issues mentioned above, the paper is clear. For the most part, the background and method are explained well. I found the math fairly straightforward to follow, and I appreciated the diagrams (Figure 2), algorithm box (Algorithm 1), and comparison overview (Table 1). The authors also provide an adequate overview of previous lines of work in the related works section (Section 4).

Some suggestions for improving the clarity of the paper:

It may help to discuss the benefits and drawbacks of parametric vs. non-parametric density estimators in more detail. Currently, the non-parametric estimator is introduced as the obvious choice, but it’s unclear why this should be any better than, for example, using a VAE or other deep generative model, as in state marginal matching [Lee et al., 2019]. When would one estimator be preferred over the other?

The intrinsic exploration reward is non-stationary, as it depends on the on-policy state distribution. It would be helpful to discuss any additional challenges associated with optimizing this reward function in comparison with task-specific reward functions.

The paper discusses entropy maximization in terms of states. Unfamiliar readers are likely more familiar with action/policy entropy maximization. It would be helpful to include a sentence or two describing the distinction between these approaches.


**Significance**: I see this paper as fairly significant for the following reasons: 1) the empirical evaluation provides results across an extensive set of environments with reasonable improvements, and 2) the method combines two fairly popular techniques in the literature, and accordingly, is likely easier to reproduce and build upon. For these reasons, I could see this paper as a useful contribution to the RL community. While this may not be the most novel or well-motivated paper, it provides a fairly clear evaluation of a useful technique.


**Minor comments**:
- Line 80: Why is T a distribution and not a density?
- Line 96: Would be helpful to explain the variable w a bit more.
- Line 131: Would be helpful to describe some of the design choices around the kernel more explicitly.
- Line 155: It’s not clear that entropy maximization is entirely modular from representation learning. For entropy maximization to work, the representation needs to contain a compressed representation of the state.
- Table 1: It’s not clear whether APT necessarily needs to be off-policy. Couldn’t one construct an on-policy version?
- Line 267: incrase —> increase
- Line 278: I’m not sure this is necessarily true for SMM.
- Table 3: Slightly unclear whether APT w/o CL refers to exploration in the pixel space or in a random encoder space.
- Line 332: phrase —> phase

---
**Update**: I have read the other reviews and the authors' responses. I had already rated this paper highly, and the authors provided a satisfactory response. The other reviewers also have increased their scores. I will maintain my score at an 8.

**Time Spent Reviewing:**

3.5

---

> ### Author Response · Authors · 2021-08-10
> **response to reviewer Zmg7**
>
> We want to thank the reviewer for the very encouraging comments. We appreciate the effort the reviewer put in providing us with constructive feedback and suggestions.
>
> 1. Would deep generative model based density modeling explore well?
>
> Using VAE and DGM based parametric models to estimate density and do maximum entropy exploration is interesting and effective in lower dimensional observation space as shown by Lee et al. However, we were not able to get such parametric exploration methods to work in Atari. Concretely, we tried the SMM with a deep neural network parameterized VAE, by checking the underlying RAM states, we found that it failed to explore even in simpler games like MsPacman, and actually worked worse than the baseline count-based exploration with PixelCNN. We believe that the difficulty of modeling density possibly explains the results.
>
> 2. Include a discussion around additional challenges associated with this intrinsic reward.
>
> As mentioned by the reviewer, one challenge is the non-stationarity of the intrinsic reward, being non additive reward poses an interesting challenge for reinforcement learning methods. We will include a further discussion around the challenges associated with maximizing this intrinsic reward in next revision
>
> 3. Discussing more about the difference between the well known policy entropy maximization and state entropy maximization.
>
> We will include a further discussion around the difference between policy entropy maximization and state space entropy maximization in the next version, thanks for the suggestion.
>
> 4. Related work in exploration.
>
> Thanks for pointing out the related work by Pitis et al, we will cite it in the next version.
>
> 5. It’s not clear whether APT necessarily needs to be off-policy. Couldn’t one construct an on-policy version?
>
> We will add a note to the comparison table to mention that APT can be either off-policy or on-policy, just in this work we opted to use off-policy for ease of optimization.
>
> 6. Language and others suggestions.
>
> We will fix all the typos and minor comments as the reviewer suggested, thank you very much for suggesting these edits.

---

### Official Review · Reviewer_zhzo · 2021-07-19

**Rating:** 7
**Confidence:** 4

**Summary:**

This paper explores the idea of doing unsupervised pre-training in a reinforcement learning context. Before giving task rewards to the agent, it is given many orders of magnitude more experience for unsupervised exploration.

The key idea is to use a non parametric entropy based reward function to efficiently explore the space and amortize it in a Q function. This Q function is then fine tuned on the task reward as a follow up stage.

Compared to previous unsupervised RL papers, this paper explores a new and interesting particle based formulation instead of count/density based model for intrinsic motivation. The performance metrics are shown on the atari and DMControl suite environments.

**Limitations And Societal Impact:**

seems adequate

**Main Review:**

The particle based entropy reward is novel as far as I understand. Typically previous work has explored the use of count based, density based and structured representations for directed exploration in a similar learning setting. However the particle based method explored here is simple, cheap to compute and asymptotically accurate. I also think that the experiment validate or answer the hypotheses considered in this paper. I also like that this paper ran strong baselines and also highlighted regimes where this approach is not as good as the baselines (table 3). So overall I think this is an interesting paper with thorough experiments and sound hypotheses.

In Table 3, VISR is much better in terms of the mean score than the proposed method. The authors explain this as -- "APT performs significantly better than VISR in hard exploration games, while VISR achieves higher scores in dense reward games". But if I look at Table 2 and the scores for pong, I would have expected VISR to perform better with this logic as it is fairly a densely rewarded environment. But this is not the case. It would be helpful to analyze and understand where, how and why VISR performs better in more detail.

 Also I am concerned about the median vs mean scores reported in the results section. How do all the results change if the number of seeds are increased or decreased?



**Time Spent Reviewing:**

2.5

---

> ### Author Response · Authors · 2021-08-10
> **response to reviewer zhzo**
>
> Thanks for taking the time to provide us with constructive feedback and valuable suggestions.
>
> 1. More analysis on where, how and why VISR performs better in some games.
>
> According to the scores in different Atari games, VISR tends to perform better APT in dense reward games, although not all dense reward games, e.g. Pong, as mentioned in the review, similarly APT performs better in most but not all exploratory games. We believe that this is due to the diversity of games in Atari benchmark, it is quite challenging to achieve better performance in every single game except Agent57 which proposes a method to combine several advances of exploration and exploitation techniques.
> We believe that understanding where, how and why VISR and APT work is an interesting and important question, there are some useful tools for analysing this e.g. Zachy et al Graying the black box: Understanding DQNs, but this might be out of scope of our work and we tend to leave it as future work.
>
> 2. Robustness of mean and median scores with respect to number of seeds.
>
> The mean and median scores are computed over 10 random seeds, which is equal or larger than the number of seeds evaluated in prior work. While increasing the number of seeds will lead to an even more statistically lower variance result, it incurs a significant amount of computational overhead. We hope that in the future we have a faster benchmark which would enable us to have a more rigorous evaluation of RL algorithms.

---

### Official Review · Reviewer_n2Z4 · 2021-07-19

**Rating:** 7
**Confidence:** 4

**Summary:**

* This paper introduces APT (Active Pre-Training), a method for unsupervised pre-training in RL environments.
* During the pertaining phase, APT explores an environment using an intrinsic reward that incentivizes maximizing entropy in a contrastive representation space.
* APT is evaluated on DM-Control and Atari to test whether pre-training improves data-efficiency. APT outperforms training RL methods from scratch in DMControl and Atari, and the gains are more noticeable in sparse reward environments.


**Limitations And Societal Impact:**

The limitations section should include the fact that APT demands a huge data budget (250M steps) for pretraining in Atari: which would be not be practical for real-world RL. It should also be pointed out APT depends on environment specific data augmentation.

**Main Review:**

This works focuses on an important research problem: How can we equip RL algorithms with strong priors about the environment so that we don’t start tabula-rasa each time we solve a task? APT approaches this by trying to learn a task-agnostic representation of an environment as well as a policy/Q-function initialization during an unsupervised pre-training phase. The representations are learned via a contrastive learning objective that leverages data augmentation, very similar to CURL, DRIML, SPR, ProtoRL and other works in this space: so this isn’t the novel aspect of this paper. To collect the data for training the encoder, APT uses an intrinsic reward that incentivizes maximizing entropy in this contrastive representation space.

The results on DMControl (particularly the sparse environments) are quite impressive, but the results on Atari are a bit underwhelming: it seems leveraging a massive amount of data (250M steps) buys little in terms of data efficiency (APT is comparable with SPR which uses no pre-training).

I have some questions about the paper, and the impact of the novel exploration method in particular which might be useful for the readers to know:

1. Since the main difference between APT and prior unsupervised RL methods is the introduction of a different intrinsic reward scheme to use for the exploration policy, it would be nice to see the comparison of how well this exploration policy performs against prior methods. In prior works like VISR, this has been well-benchmarked: see Table 1 (first section) of [VISR](https://arxiv.org/abs/1906.05030) [1] which reports Curiosity@0 and VISR@0. Comparing the APT@0 scores would show whether APT performs better exploration than prior methods during the unsupervised phase.

2. I am concerned that contrastive learning is going to lead to maximally spread-out representations that naturally maximize APT’s exploration reward without any actual exploration at all.  A core part of contrastive learning is the "uniformity on the hypersphere" [2] property, which means that points in the replay buffer should be encoded as maximally distant from other points in the replay buffer. Hence, it’s not clear that APT’s exploration is much more sophisticated than giving the agent a reward for staying alive.  Note that the VISR paper does report this baseline:
>To rule this out, an explicit “do not die” baseline was run (Pos Reward NSQ), wherein the terminal signal remains and a small constant reward is given at every time-step.

So, if we compare with APT@0 with “Pos Reward NSQ @0” (Table 1 in VISR), we should be able to disentangle if this is actually the case.

3. The ablations section is limited and I want to better understand which components of APT provide a strong learning prior, and which don’t. Table 3 and Table 4 already shows that CL is indeed quite helpful, but I am not sure how much do the learned policy/q-networks contribute. Do you have an experiment that ablates  the performance of a randomly initialized policy/q-network vs their pre-trained counterparts?

4. The most interesting experiments in the paper are on the sparse reward environments in DMControl, showing a remarkable improvement in both data efficiency and final performance on sparse environments. I was a bit confused to find out though that these sparse rewards were custom designed, and not evaluated on any of the standard sparse environments already available in DMControl (Cartpole Swingup Sparse, Cartpole Balance Sparse, etc.). Introducing both new envs and new methods in the same paper can often lead to selection bias, and it would be nice to see if the results on already existing DMControl envs so that future works also can benchmark against these results easily. See for example ProtoRL [3], another unsupervised RL method that uses these sparse envs.

5. Can you clarify what "Value of k" = "search in {3, 5, 10}" in the supplementary refers to? Does this mean the value of k was tuned for each env separately. If so, this is problematic, and deviation from the standard evaluation protocol in Atari where the same algo with the same hyperparams are used for all games. This also makes comparisons to baseline unfair since none of the baselines used per-env hyper-param tuning.

Missing prior work: Though the paper in general does a good job of covering prior work, it misses ProtoRL which is a relatively/concurrent work, and so its okay to not directly compare to it but it should be cited nonetheless.  There should also be a mention of Plan2Exlpore in the related work section.

Minor comments on the writing:

1. In Algo 1: “Train neural encoder f on mini batch” is vague: you can reference the contrastive learning equation here. There should also be one more step that clarifies that data augmentation is performed on the mini-batch. Instead of standard Q-learning, you should say standard actor-critic (since both policy and Q-networks are mentioned).
2. "With the models and datasets growing, there is still no sign of saturating performance during pre-training." The scaling law curves do start to saturate after reaching irreducible entropy, perhaps a better way to phrase this is “With increasing dataset and mode sizes, performance continues to improve predictably according to scaling laws.”
3. The imagenet pertaining experiment in Figure 1 has no details, and in-fact a recent paper ([2107.03380 RRL: Resnet as representation for Reinforcement Learning](https://arxiv.org/abs/2107.03380)) contradicts these results (the paper was released after the NeurIPS deadline). It would be nice to have details of the ImageNet experiment in the supplementary material so that we could compare the differences in experimental setup between these two.

[1] Hansen, Steven, Will Dabney, Andre Barreto, Tom Van de Wiele, David Warde-Farley, and Volodymyr Mnih. "Fast task inference with variational intrinsic successor features." ICLR 2020

[2] Wang, Tongzhou, and Phillip Isola. "Understanding contrastive representation learning through alignment and uniformity on the hypersphere." ICML 2020

[3] Yarats, Denis, Rob Fergus, Alessandro Lazaric, and Lerrel Pinto. "Reinforcement learning with prototypical representations." ICML 2021

**Time Spent Reviewing:**

6

---

> ### Author Response · Authors · 2021-08-10
> **response to reviewer n2Z4**
>
> We want to thank the reviewer for all the great suggestions and valuable feedback, we appreciate the effort and time the reviewer put in.
>
> 1. Compared with SPR the gain is not significant.
>
> SPR is a state of the art model-based algorithm, while its representation is also learned by contrastive learning (byol), it trains a model-based dynamic to ​​predict its own latent state representations multiple steps into the future. This temporal representation learning, as illustrated in SPR paper, contributes to its impressive results compared with standard contrastive representation learning. We believe that it is possible to combine this temporal contrastive representation learning of SPR with the effective nonparametric entropy maximization of SPR, which is worth studying as future work.
>
>
> 2. Investigate APT@0 and “Pos Reward APT @0”.
>
> To resolve the concern around whether maximum entropy exploration is needed given that contrastive learning has the “uniformity on the hypersphere”[2] property. Similar to using a do not die signal in “Pos Reward NSQ” as mentioned by the reviewer, we ran experiments with “APT do not die” which receives only positive do not die signals and keeps representation learning the same. We ran the experiments in MsPacman, due to limited resources, we reduced the pretraining phase to 5M steps. The evaluation metrics are the number of ram states visited and the downstream zero shot performance. APT visits nearly 27 times more unique ram states than “Pos Reward APT”, showing that the entropy intrinsic reward is indispensable for exploration. In downstream task evaluation over 3 random seeds, “Pos Reward APT@0” achieves reward 363.7, “APT@0” achieves reward 687.1, showing that the “do not die” signal is insufficient for exploration and learning representation.
>
>
> 3. Randomly initialized policy/q-network vs their pre-trained counterparts.
>
> Our preliminary experiments showed that finetuning from a pretrained actor-critic head accelerates learning. We have run experiments in five different Atari games, as shown in the table below, pretrained heads perform better than randomly initialized heads in 4 out of 5 games. We believe that which one of the two is better depends on the alignment between downstream reward and intrinsic reward. It would be interesting to study how to better leverage downstream reward to finetune the pretrained model.
>
> | Mean Reward (3 seeds)      | Alien | Freeway | Qbert | Private Eye | MsPacman |
> | :---        |    :----:   |         : ---: |       :----:   |          :---: |          :---: |
> | APT (pretrained head)      | 2614.8 | 29.9 | 17671.2 | 96.1 | 2827.1 |
> | APT (random head)   | 1755.0 | 15.2 | 2138.3 | 61.3 | 1724.9 |
>
>
> 4. Some sparse reward tasks evaluated are designed by the authors.
>
> The reason we opted to design new sparse reward tasks rather than using the existing CartPole is to have more diverse downstream tasks. As far as we know, there is only one Cartpole Swingup Sparse that is a CartPole based sparse reward task. Due to its 2D nature being quite limited, we eventually decided to design distinguishable downstream tasks based on more complex environments, e.g. Hopper Jump etc. We will include the full xml configuration files and details of our customized tasks in the updated version to facilitate future research.
>
>
> 5. The results of ImageNet pretraining experiments.
>
> While RRL shows freezed ImageNet pretrained resnet is helpful, other prior work including the [Never Stop Learning Julian et al](https://arxiv.org/pdf/2004.10190.pdf) show that ImageNet's pretrained encoder achieves zero-success rate and only with finetuning it achieves non-zero success rates.
> In our setup, we observed the similar negative results, using a freezed pretrained ImageNet model has suboptimal performance and with finetuning it can match randomly initialized encoder. We tried using publicly available ImageNet pretrained ResNet50 with upsampling for our tasks, we also tried training different variants including ResNet50, ResNet18, with and without downsampling in ImageNet and using the pretrained models for our tasks.
> Here is a summary of the conclusions of NSL, RRL, and APT.
>
> - NSL: freezed imagenet pretrained encoder achieves zero success rate in real world robotics , with finetuning the pretrained model improves performance.
> - RRL: using a freezed pretrained ImageNet encoder can improve success rate in simulated robot manipulation.
> - APT: freezed encoder has significantly worse asymptotic performance, with finetuning it has similar results as randomly initialized encoder in simulated continuous control.
>
> Due to all the difference in task setup and implementation details, we unfortunately don’t have a definite answer regarding the usefulness of ImageNet pretrained encoder.
> It would be clearer if RRL includes an ablation study using a freezed randomly initialized encoder, the majority of performance gain might be due to the architecture of resnet rather than pretrained weights.
> In the Figure 2 of RRL, the visualized attention focuses on the robot hand and object, but it’s unclear whether a randomly initialized resnet can achieve similar results especially since deep networks tend to focus on the bright pixels in an image.
> Taking the ViT for example (which is easier to visualize attention than convnet), a randomly initialized ViT can still show reasonable results in real images. To see this, just run the attention visualization python script in DINO github codebase using an untrained model in ImageNet images or MS COCO images, happy to share some results if the reviewer is interested in.
>
> We agree with the reviewer that more experimental details about our ImageNet pretraining should be included in the paper, we will include the all the details of our experiments in our updated version and release a IPython notebook for finetuning ImageNet pretrained model before the conference.
>
>
> 6. Language and others suggestions.
>
> We will fix all the typos and minor comments as the reviewer suggested, thank you very much for suggesting these edits.

---

> > ### Comment · Reviewer_n2Z4 · 2021-08-23
> > **Post rebuttal**
> >
> > Thanks for the rebuttal and addressing most of my concerns, I am updating my rating to a 7.
> >
> > I hope the additional experiments and details make it to the next version of the paper. I would also recommend expanding on the preliminary APT@0 experiments to cover more games..

---

### Official Review · Reviewer_D1C9 · 2021-07-20

**Rating:** 6
**Confidence:** 5

**Summary:**

The authors introduce Active Pretraining (APT), wherein an agent is encouraged to maximization the entropy of the state distribution represented in its replay buffer. The is accomplished by having rewards that are proportional to distance to nearest neighbours in some embedding space. This space is constructed via a separate contrastive learning objective.

This method is tested in a 2 stage process where the agent initially maximizes this state entropy entropy for a long time, and then finetunes on a down-stream time for a much shorter time, with the final performance on this task being the key metric. By this methodology, APT surpasses both data-efficient methods that train from scratch, as well as other methods that use the same 2 stage training scheme.

**Limitations And Societal Impact:**

yes

**Main Review:**

The novelty of the method is a bit slight, as using state-entropy as an intrinsic reward has been attempted elsewhere. That said, the authors acknowledge this prior work, and emphasize the scalability and performance of their system. This puts much more pressure on the empirical results to do the heavy lifting, and luckily they largely deliver on this promise.

The initial negative result showing that naive approaches to pretraining (e.g. ImageNet) are insufficient to accelerate downstream tasks provides some nice motivation. The main results on DMControlSuite and Atari have a large number of baselines whose inclusion is well motivated in the text. The performance of APT is quite convincing, but I would suggest including some notion of spread in all of the table and figures so that the significance is easier to assess. Both ablating contrastive learning from APT and including it in some baseline methods was a nice touch, and the other ablations are also appreciated. That said, Appendix A did give me pause, as the number of architectural flourishes used here feels higher than many other RL papers (e.g. layer norm, spectral norm), but considering you compare favourably to a Rainbow variant (which is probably the fanciest data efficient agent to use as a base) I won't hold that against you.

One significant concern involve proposition 1. It's correct, if a bit trivial, but it's unclear that it matters in practice, as the number of distinct states is far higher than the number of training steps for the pixel-based domains that are core to this method's evaluation. The applicability is further undercut by the reward normalization scheme -- as the reward goes to zero, the normalized reward goes to infinity! And I'm not even convinced one would even want the intrinsic reward to vanish asymptotically, as that suggests that pretraining for too long would result in a random policy. Luckily I think these two problems cancel out; if you don't want your reward to vanish, then just removing the proposition and the surrounding text would alleviate my concern.

Minor points:
1) The "Exploration" tick marks in Table 1 feel very subjective. Maximizing state or action entropy is exploration, but related mutual information terms isn't? Even if there is an argument for this distinction, since just switching to a base RL method that includes an entropy bonus (e.g. SAC) would resolve it, categorizing methods along this axis feels unhelpful.
2) The fine-tuning strategy employed here (when combined with a value function) relies on the intrinsic and extrinsic rewards being of a similar scale. This is in contrast to the regression-style approach of VISR, and should thus probably be mentioned as a weakness of APT unless its already being addressed by other means (e.g. van Hasselt et al).

van Hasselt, Hado P., et al. "Learning values across many orders of magnitude." Advances in Neural Information Processing Systems 29 (2016): 4287-4295.


**Time Spent Reviewing:**

3

---

> ### Author Response · Authors · 2021-08-10
> **response to reviewer D1C9**
>
> We want to thank the reviewer for the constructive comments and suggestions.
>
> 1. Statement that MI based methods have insufficient exploration.
>
> We will revise the statement to make it more clear, what we intend to say is that most MI based methods do not have sufficient exploration. The reason is that to make MI objective tractable, the MI I(w;s) is decomposed as H(w) - H(w|s) where w is a random variable sampled from a fixed distribution, and a variational lower bound of -H(w|s) is used to derive intrinsic reward. This intrinsic reward encourages RL agents to predict w from induced states but does not encourage exploration. This issue could be partially alleviated by deploying entropy regularized RL, e.g. SAC, as mentioned in the review, however, intrinsic reward with explicit exploration properties such as count based bonus trained using DrQ (SAC + data augmentation) does not work well in pixel domains. We believe that using entropy regularized RL can possibly improve exploration of MI based methods but it is difficult to match or outperform other exploration oriented intrinsic rewards.
> There are a line of work in decomposing MI into the reverse direction as H(s) - H(s|w), such as in Campos et al. Explore, Discover and Learn: Unsupervised Discovery of State-Covering Skills, this objective has the property of explicitly exploration and combining our nonparametric entropy maximization with MI based methods is an interesting work.
>
> 2. Normalization by a running estimate of the mean of the intrinsic reward and vanishing property.
>
> As mentioned by the reviewer, dividing a running estimate of the mean may conflict with vanishing reward. While empirically this is not an issue, due to even without normalization letting the reward vanish needs a nearly impractical number of training steps.
> We agree that vanishing reward is not necessarily a desired property, although it could serve a signal of completing pretraining.
> Regarding normalization by a running estimate of the mean, it is used to make the intrinsic reward more robust to the task being solved, as different tasks may have different typical distances between latent representations. In our preliminary experiments, we found that doing so leads to slightly more robust results across different games.
>
> 3. Different scales between intrinsic reward and downstream reward.
>
> Thanks for suggesting van Hasselt et al, we believe it is an interesting future direction to combine the method with APT and finetune for multi-task with different reward scales. We believe the discrepancy between intrinsic reward scale and downstream reward scale possibly explain the suboptimal performance of APT in dense reward games. This is an interesting future direction to further improve APT, we hypothesize that reinitializing behaviors part (actor critic heads) might be useful if the downstream reward scale is very different from pretraining reward scale. A more principled way would be adaptive normalization as in van Hasselt et al, it is definitely an interesting future direction.
>
> 4. Language and others suggestions.
>
> We will fix all the typos and minor comments as the reviewer suggested, thank you very much for suggesting these edits.

---

> > ### Comment · Reviewer_D1C9 · 2021-08-23
> > **Proposition 1**
> >
> > Thank you for your thoughtful rebuttal -- I particularly appreciate the additional clarity being added wrt exploration in MI methods.
> >
> > That said, I'm still a bit concerned about proposition 1
> >
> > > As mentioned by the reviewer, dividing a running estimate of the mean may conflict with vanishing reward. While empirically this is not an issue, due to even without normalization letting the reward vanish needs a nearly impractical number of training steps. We agree that vanishing reward is not necessarily a desired property, although it could serve a signal of completing pretraining. Regarding normalization by a running estimate of the mean, it is used to make the intrinsic reward more robust to the task being solved, as different tasks may have different typical distances between latent representations. In our preliminary experiments, we found that doing so leads to slightly more robust results across different games.
> >
> > I agree with this statement, but to my mind this further enforces the idea that proposition 1 isn't particularly meaningful -- your algorithm doesn't meet its conditions and even if it did it wouldn't matter in practice. If proposition 1 is removed from the final draft of the paper I will no longer have any significant issues with the papers and will raise my score to a 7.

---

> > > ### Author Response · Authors · 2021-08-26
> > > **reply to reviewer D1C9**
> > >
> > > Thank you for the suggestion regarding proposition 1, we will remove it in the next version.

---

### Decision · Program_Chairs · 2021-09-27

**Decision:**

Accept (Spotlight)

**Comment:**

This paper proposes a novel unsupervised pre-training method for reinforcement learning, using a particle-based entropy estimator on a contrastive-loss trained feature representation. The method is novel and scalable and the empirical results and rebuttal additional experiments are strong. I trust the authors will incorporate the reviewers' comments and discussion points into the final version of the manuscript.